# FGF21 Levels in Pheochromocytoma/Functional Paraganglioma

**DOI:** 10.3390/cancers11040485

**Published:** 2019-04-05

**Authors:** Judita Klímová, Tomáš Zelinka, Ján Rosa, Branislav Štrauch, Denisa Haluzíková, Martin Haluzík, Robert Holaj, Zuzana Krátká, Jan Kvasnička, Viktorie Ďurovcová, Martin Matoulek, Květoslav Novák, David Michalský, Jiří Widimský Jr., Ondřej Petrák

**Affiliations:** 1Third Department of Medicine, Department of Endocrinology and Metabolism of the First Faculty of Medicine, Charles University and General University Hospital in Prague, 128 00 Prague, Czech Republic; Tomas.Zelinka@vfn.cz (T.Z.); Jan.Rosa@vfn.cz (J.R.); Branislav.Strauch@vfn.cz (B.Š.); Robert.Holaj@vfn.cz (R.H.); Zuzana.Kratka@vfn.cz (Z.K.); Jan.Kvasnicka3@vfn.cz (J.K.); viktoria.durovcova@gmail.com (V.Ď.); martin.matoulek@vstj.cz (M.M.); Jiri.Widimsky@vfn.cz (J.W.J.); Ondrej.Petrak@vfn.cz (O.P.); 2Institute of Sports Medicine, First Faculty of Medicine, Charles University and General University Hospital in Prague, 120 00 Prague, Czech Republic; Denisa.Haluzikova@vfn.cz; 3Centre for Experimental Medicine and Diabetes Centre, Institute for Clinical and Experimental Medicine, 140 21 Prague, Czech Republic; martin.haluzik@ikem.cz; 4Institute for Medical Biochemistry and Laboratory Diagnostics, Charles University and General University Hospital in Prague, 128 08 Prague, Czech Republic; 5Department of Urology of the First Faculty of Medicine, Charles University and General University Hospital in Prague, 128 00 Prague, Czech Republic; Kvetoslav.Novak@vfn.cz; 6First Department of Surgery of the First Faculty of Medicine, Charles University and General University Hospital in Prague, 128 00 Prague, Czech Republic; david.michalsky@vfn.cz

**Keywords:** FGF21, pheochromocytoma, paraganglioma, diabetes mellitus, obesity, energy metabolism, calorimetry

## Abstract

Fibroblast growth factor 21 (FGF21) is a hepatokine with beneficial effects on metabolism. Our aim was to evaluate the relationship between the serum FGF21, and energy and glucose metabolism in 40 patients with pheochromocytoma/functional paraganglioma (PPGL), in comparison with 21 obese patients and 26 lean healthy controls. 27 patients with PPGL were examined one year after tumor removal. Basic anthropometric and biochemical measurements were done. Energy metabolism was measured by indirect calorimetry (Vmax-Encore 29N). FGF21 was measured by ELISA. FGF21 was higher in PPGL than in controls (174.2 (283) pg/mL vs. 107.9 (116) pg/mL; *p* < 0.001) and comparable with obese (174.2 (283) pg/mL vs. 160.4 (180); *p* = NS). After tumor removal, FGF21 decreased (176.4 (284) pg/mL vs. 131.3 (225) pg/mL; *p* < 0.001). Higher levels of FGF21 were expressed, particularly in patients with diabetes. FGF21 positively correlated in PPGL with age (*p* = 0.005), BMI (*p* = 0.028), glycemia (*p* = 0.002), and glycated hemoglobin (*p* = 0.014). In conclusion, long-term catecholamine overproduction in PPGL leads to the elevation in serum FGF21, especially in patients with secondary diabetes. FGF21 levels were comparable between obese and PPGL patients, despite different anthropometric indices. We did not find a relationship between FGF21 and hypermetabolism in PPGL. Tumor removal led to the normalization of FGF21 and the other metabolic abnormalities.

## 1. Introduction

Fibroblast growth factor 21 (FGF21) is a metabolic regulator that has a systemic effect in promoting glucose uptake and oxidation [1,2]. The main site of FGF21 expression and production in humans is liver and, to a lesser degree, muscle and white adipose tissue (WAT) [3]. In rodents, FGF21 targets brown adipose tissue (BAT), where it induces mitochondrial uncoupling protein-1 (UCP1) gene expression and favors glucose oxidation and energy expenditure [4,5].

In humans, under conditions of enhanced adaptive energy expenditure, brown adipocyte-like cells appear at sites of WAT. This is called the “browning” of WAT, and cells resembling brown adipocytes arising in this process are called “beige” [6]. The promotion of WAT browning was documented in patients with pheochromocytoma and functional paragangliomas (PPGL) due to the tumor-mediated release of catecholamines [7,8]. Analysis of visceral adipose tissue from the perirenal and omental regions in small samples of pheochromocytoma confirmed the presence of beige adipose tissue with significant expression of FGF21 [9].

Furthermore, there is also an adipose-independent mechanism for FGF21 to be able to regulate metabolism [10,11]. FGF21 can cross the blood-brain barrier and is detectable in both human and rodent cerebrospinal fluid [12,13]. Continuous intracerebroventricular injection of FGF21 to obese rats increases energy expenditure and insulin sensitivity [14]. FGF21 has been shown to act on the central nervous system to increase systemic glucocorticoid levels, suppress physical activity, and alter circadian behavior [15,16].

PPGL represents a useful model for studying the influence of long-term catecholamine overproduction in metabolic disorders in humans. Catecholamine overproduction leads to a large variety of signs and symptoms, including sustained or paroxysmal arterial hypertension, hypermetabolic state with weight loss and disorders of glucose metabolism [17]. The aim of our study was to evaluate the changes of circulating levels of FGF21 and its relationship to energy and glucose metabolism in PPGL, before and one year after tumor removal. For comparison, at baseline, we used healthy lean controls and also obese patients with glucose metabolism disorder.

## 2. Results

### 2.1. Basic Characteristics of Groups

Biochemical, anthropometrical, and clinical characteristics of the studied groups are summarized in Table 1. Obese patients and controls contained significantly more females than PPGL (*p* = 0.008). As expected, obese patients had a higher body mass index (BMI) and body fat percentage in comparison with PPGL and controls (*p* < 0.001). Also their lipid profile differed in triglycerides (TAG) and high-density lipoprotein cholesterol (HDLc) (*p* < 0.001). Higher values of resting energy expenditure/basal energy expenditure (REE/BEE) were present in PPGL and obese in comparison with controls. Hypermetabolic state was detected in 49% of PPGL and 24% of obese. PPGL and obese also showed higher systolic blood pressure (sBP) than controls (*p* = 0.002). Fasting blood glucose (FBG) and glycated hemoglobin (HbA1c) levels in PPGL and obese were similar (*p* = NS), but higher than in controls (*p* < 0.001). Insulin levels were lower in PPGL patients than in controls (*p* = 0.037) and expectedly higher in obese, together with HOMA-IR (*p* < 0.001). Adrenergic phenotype was seen in 61% of PPGL patients and noradrenergic phenotype in 39% of PPGL patients.

### 2.2. Effect of Tumor Removal in PPGL

A subgroup of 27 consecutive patients with PPGL was examined one year after tumor removal. The basic characteristics of patients before and after tumor removal are summarized in Table 2. The decrease in free plasma metanephrines and chromogranine reflects successful surgery (*p* < 0.001). Weight gain was significant in all parameters, including body fat percentage. No patient fulfilled the requirements for hypermetabolism after tumor removal. No significant decrease in office blood pressure measurements was present. Sustained hypertension remained in 22% of patients, but the total number of antihypertensives decreased. FBG and HbA1c decreased significantly. Unexpectedly, insulin levels did not normalize. On the contrary, normal glucose tolerance was present in the majority of patients (93%).

### 2.3. FGF21 Levels and Correlation

FGF21 levels were broad in all groups. The correlation between FGF21 and other selected factors in patients with PPGL is shown in Table 3 and Figure 1. FGF21 was significantly higher in PPGL than in controls (*p* < 0.001) and comparable with obese (*p* = NS). PPGL patients with diabetes showed higher levels of FGF21 than those with normal glucose tolerance (NGT) (438.2 (337) pg/mL vs. 154.5 (97) pg/mL; *p* = 0.007) or those with prediabetes (438.2 (337) pg/mL vs. 154.5 (97) pg/mL; *p* = 0.022). Diabetic obese and obese with NGT showed similar results (314.1 (300) pg/mL vs. 140.5 (7) pg/mL; *p* = 0.049). Similar differences in FGF21 levels were present in diabetic and prediabetic obese (314.1 (300) pg/mL vs. 113.1 (54) pg/mL; *p* = 0.024). Those findings are summarized in Figure 2. PPGL with metabolic syndrome or dyslipidemia showed higher FGF21 than those without (*p* < 0.001). FGF21 levels differed in PPGL with adrenergic and noradrenergic phenotypes, but were slightly above the level of significance (*p* = 0.062]. A difference in FGF21 levels between hypermetabolic and normometabolic PPGL patients was not found (*p* = NS). FGF21 levels in all components of metabolic syndrome in PPGL are shown in Table 4.

## 3. Discussion

Our study shows that patients with PPGL have higher serum levels of FGF21 compared to healthy controls and these levels do not differ from obese patients. Furthermore, successful tumor removal significantly decreased FGF21 levels. Elevated FGF21 levels were more evident in patients with secondary diabetes mellitus and were related positively to fasting glucose levels and BMI in PPGL. We did not find a relationship between FGF21 and hypermetabolic state in PPGL.

We know from animal models that FGF21 stimulates whole-body energy expenditure and increases metabolic rate and physical activity [18]. In humans, conflicting results were published, depending on the studied population. In healthy lean volunteers, augmented FGF21 levels correlated positively with total energy expenditure during cold exposure [19]. In another study, fasting and postclamp FGF21 were positively related to REE, particularly in obese subjects [19]. On the other hand, no association between FGF21 and REE was reported in patients with hypercortisolism and a healthy population with low birth weight [20,21].

Catecholamine overproduction in PPGL leads to an increase in resting energy expenditure [22]. Although there are elevated levels of FGF21 in patients with PPGL, we have not found a link to hypermetabolic state. The causes of the hypermetabolic condition are likely to be much more complex and include the direct action of catecholamines on intermediate metabolism, fatty tissue, and inflammation [22,23,24]. In a fat biopsy study in PPGL, increased activity of brown adipose tissue in visceral fat, along with mRNA FGF21 was demonstrated, compared to patients undergoing elective cholecystectomy [9]. However, the study did not compare serum levels of FGF21. Thus we speculate that serum levels of FGF-12 may not reflect local paracrine production and activity in adipose tissue in PPGL. Another explanation can be found in the study by Douris and co-workers. They demonstrated that FGF21 also acts centrally in the brain through the activation of the sympathetic nervous system, which induces browning of WAT [11]. They also demonstrated that an intact beta-adrenergic receptor signaling pathway is necessary for the central actions of FGF21 [11]. It was shown that chronic overproduction of catecholamines could lead to desensitization of beta-adrenergic receptors in PPGL [25,26]. These findings and the existence of genetic polymorphisms of the beta-adrenergic receptor lead us back to actions (both, catecholamines and FGF21) on local levels.

Experimental studies have shown that it is noradrenaline which stimulates the production of FGF21 in brown adipose tissue, via beta-adrenoceptors [5]. In addition, this effect is not affected by the concomitant administration of an alpha-blocker [5]. Surprisingly, we did not find a link either between FGF21 and both free plasmatic metanephrines or noradrenergic and adrenergic biochemical phenotype. In our study, we used free plasmatic metanephrines, which are the gold standard for the biochemical diagnosis of PPGL. They are produced continuously within PPGL tumor cells, and independently of catecholamine release, and they do not reflect the biochemical activity of tumors. We assume that this could explain the weak link between noradrenergic phenotype and serum FGF21 levels in our study.

Mraz and co-workers have demonstrated that FGF21 expression in the human liver was more than 100-fold higher relative to fat, suggesting that the liver remains the most important producer of this factor in humans [27]. Enhanced liver production of FGF21 has been linked to obesity, diabetes mellitus, and metabolic syndrome [27,28,29]. Our study reveals the same finding. Obese patients with metabolic syndrome had a higher level of FGF21, which correlates with blood glucose and BMI levels. In patients with PPGL, the findings were similar, despite significantly lower BMI. However, circulating levels of FGF21 were significantly higher in PPGL with secondary diabetes mellitus and signs of metabolic syndrome. The question remains whether it is the presence of hyperglycemia that stimulates the production of FGF21 in PPGL. According to available studies, human serum FGF21 levels are increased by oral boluses of fructose and glucose [30,31,32] and by 24-h hyperglycemia maintained via intravenous glucose infusion [33]. Furthermore, von Holstein-Rathlou et al. demonstrated in cell culture (HepG2) and mice models that glucose and fructose directly influence hepatocytes in the production of FGF21 [33]. On the other hand, Samms et al. showed that insulin rather than glucose “per se” increases total and bioactive FGF21 in the postprandial period in adult humans with and without type 2 DM according to oGTT and glucose clamp [34]. In our group with PPGL, basal insulin levels were significantly lower than in the obese and lean controls, and we did not find a relation to FGF21. It is possible that the mechanism of the insulin resistance state in PPGL is different. Insulin secretion is impaired, due to the inhibitory effect of catecholamines by the activation of α-adrenergic receptors in pancreatic β cells. In addition, catecholamines antagonize insulin action in target organs and thereby might trigger insulin resistance [35,36,37]. Komada et al. found impairment of insulin secretion particularly in the early phase of the insulin secretory response [38]. Our work shows that metabolic changes in PPGL are partially reversible. One year after adrenalectomy, we find an improvement in glucose metabolism and insulin resistance, followed by a decrease in FGF21, despite an increase in body weight due to the disappearance of the hypermetabolic effect of catecholamines.

The consequences of the elevation of FGF21 in PPGL are unclear. We cannot identify from our work whether the elevation of FGF21 is the result of a controversial “FGF21 resistance state”, as is known in obese and diabetic patients [39,40,41], or whether FGF21 has some biological effect. However, in comparison to serum FGF21 levels in PPGL versus obese individuals, the levels in PPGL are most likely biologically significant. From the context mentioned above, we found that circulating FGF21 in PPGL reflects the metabolic abnormalities associated with diabetes mellitus and metabolic syndrome components, and we did not find a relation to hypermetabolism. Thus, it is possible that circulating levels of FGF21 originate predominantly from hepatic production, as demonstrated in the mice model [42]. Further investigation would be needed to assess the effect of FGF21 on metabolism and adipose tissue in PPGL.

Our study has several limitations. Firstly, the range of FGF21 serum concentration in human studies is very wide, making interpretation of clinical observations difficult. Secondly, our population was small and of a cross-sectional nature. Thirdly, we cannot exclude the influence of antihypertensive or antidiabetic therapy in both treated groups. Fourthly, we measured only total FGF21 and not the bioactive form of FGF21 and other important proteins such as fibroblast- activating protein. Finally, the lack of determination of urinary catecholamine levels to assess metabolic effects is another limitation of our study.

## 4. Patients and Methods

### 4.1. Recruitment and Background

87 subjects were included in the study (40 patients with PPGL, 26 healthy volunteers, and 21 obese individuals). Subjects with PPGL (38 pheochromocytoma and 2 abdominal paraganglioma) were examined during a short hospitalization in our department before and one year after tumor removal. Diagnosis of PPGL was based on free plasma metanephrines levels, visualization of the tumor by computer tomography or PET/CT with fluorodopa. The diagnosis was confirmed histopathologically. Five patients with familial, bilateral, or malignant PPGL were not examined after operation. Obese participants were investigated during hospitalization for weight reduction. All obese patients had hypercortisolism excluded. Healthy subjects had no history of chronic disease or medication. Written informed consent was obtained from all patients. The ethical committee of our institution approved the study (permission date: 21 May 2015, ethical code: 20/15). The study was done in accordance with the Declaration of Helsinki.

### 4.2. Anthropometric, Biochemical Measurements, and Indirect Calorimetry

Blood samples were withdrawn after overnight fasting between 6 and 7 a.m. Height (cm), weight (kg), and waist and hip circumference (cm) were measured. BMI was calculated as weight in kilograms divided by the square of height in meters. Obesity was defined by BMI >30 kg/m². Metabolic syndrome was classified according to the International Diabetes Federation by the presence of central obesity (BMI >30 kg/m² or waist circumference ≥94 cm in males or ≥80 cm in females), and any two of the following four factors: triglycerides ≥1.7 mmol/L or specific treatment for these lipid abnormalities; HDL cholesterol <1.03 mmol/L in males and <1.29 mmol/L in females or specific treatment for these lipid abnormalities; systolic blood pressure ≥130 or diastolic blood pressure ≥85 mmHg or treatment of previously diagnosed hypertension; fasting blood glucose ≥5.6 mmol/L or previously diagnosed type 2 diabetes.

Office arterial blood pressure was measured with an oscillometric sphygmomanometer according to the European Society of Hypertension guidelines. Arterial hypertension was defined in accordance with the European Society of Hypertension guidelines.

Basic laboratory tests, including serum glucose, lipid profile, glycated hemoglobin (HbA1c), were measured by standard methods in our institutional laboratory with international accreditation. Insulin was analyzed by the IRMA kit BI-INS-IRMA (Cis Bio International, Sark France). Homeostasis model assessment index-insulin resistance (HOMA-IR) was calculated as fasting glucose concentration multiplied by fasting insulin and divided by 22.5. Diabetes mellitus was defined by fasting plasma glucose levels ≥7.0 mmol/L or plasma glucose ≥11.1 mmol/L two hours after a 75 g oral glucose load or HbA1C ≥48 mmol/mol. Prediabetes was defined by fasting blood glucose levels from 5.6 to 6.9 mmol/L or plasma glucose ≥7.8 mmol/L, but not over 11.1 mmol/L, two hours after a 75 g oral glucose load according to the WHO 2006 definition.

Serum FGF21 levels were measured by a commercial ELISA kit (BioVendor, Modrice, Czech Republic), which is based on the polyclonal anti-human FGF21 antibody and biotin-labeled polyclonal anti-human FGF21 antibody. Plasma free metanephrines (normetanephrine and metanephrine) were quantified by liquid chromatography with electrochemical detection (HLPC-ED, Agilent 1100, Agilent Technologies, Inc., Wilmington, DE, USA). The noradrenergic biochemical phenotype was defined as: predominant increases of only normetanephrine, accompanied by either normal plasma concentrations of metanephrine or by increases of less than 5 % for metanephrine relative to the sum of increments for both hormones. The adrenergic biochemical phenotype was defined as: increases of plasma metanephrine above the upper reference limits and relative to the combined increments of both metabolites, of larger than 5 % for metanephrine [43].

Energy metabolism was quantified by indirect calorimetry with a ventilated canopy system (Vmax Encore 29 N system, VIASYS Healthcare Inc; SensorMedics, Yorba Linda, California). Resting energy expenditure (REE) and respiratory quotient (RQ) were measured. The methodology was described in our previous article [22]. The Harris-Benedict formula was used for the calculation of predicted basal energy expenditure (BEE). REE was divided by BEE and multiplied by 100 to express the rate of metabolism (REE/BEE). Hypermetabolic state was classified as REE/BEE more than 110%. Free fat mass (FFM) was measured with a Bodystat 1500 device (Bodystat Ltd, Isle of Man, UK).

### 4.3. Statistical Analysis

Statistical analysis was performed by Statistica for Windows ver. 9.1 (StatSoft, Inc., Tulsa, OK, USA). Normally distributed data are shown as the mean ± SD (standard deviation). Data with abnormal distribution are expressed as median with interquartile range (IQR). Categorical variables are expressed as frequencies (%). All parameters were tested for normality by the Shapiro-Wilk test. Parameters without a normal distribution were logarithmically converted. Two independent groups were tested by the Student’s *t*-test or Mann-Whitney test as appropriate. The dependent groups were tested by the Student’s paired *t*-test or the Wilcoxon test as appropriate. For three and more groups, the Kruskal-Wallis test, or an ANOVA with Scheffe post-hoc test, was used. Correlations between variables were investigated by the Pearson correlation coefficient. Categorical variables were tested by chi-square or Fisher’s exact test. *p* values of <0.05 were considered significant.

## 5. Conclusions

In conclusion, we found elevated levels of serum FGF21 levels in PPGL and their relation to secondary diabetes mellitus, but not to the hypermetabolic state. One year after tumor removal led to normalization of FGF21 and the other metabolic abnormalities.

## Figures and Tables

**Figure 1 cancers-11-00485-f001:**
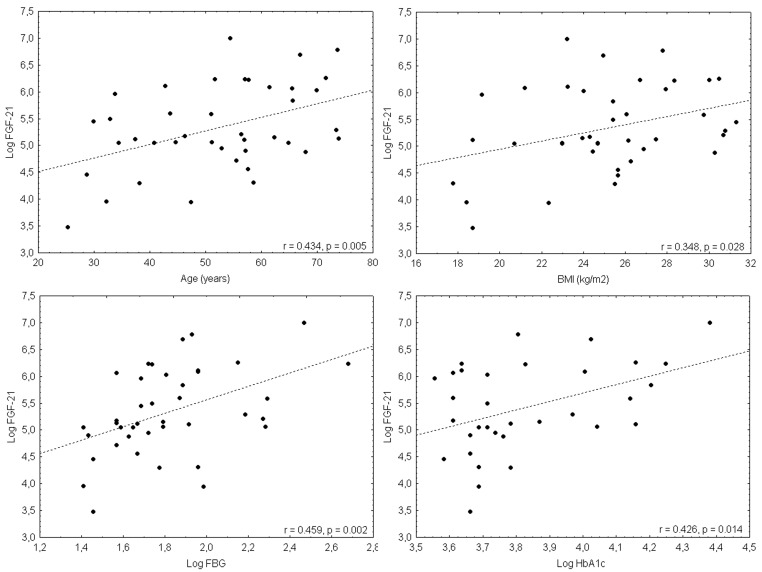
FGF21 levels and their correlation with selected factors in PPGL—age, BMI, FBG a HbA1c.

**Figure 2 cancers-11-00485-f002:**
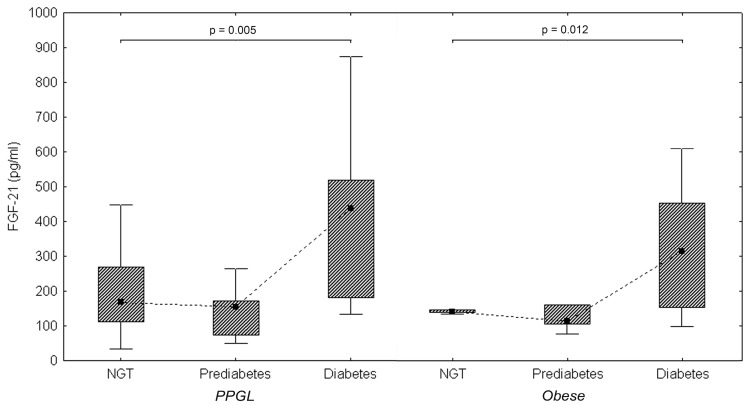
FGF21 levels in obese and PPGL with normal glucose tolerance (NGT), prediabetes, and diabetes mellitus.

**Table 1 cancers-11-00485-t001:** Clinical and metabolic characteristics of study subjects.

Factors	Controls *n* = 26	Obese *n* = 21	PPGL *n* = 40	*p*
Female (*n*, %)	21 (81)	18 (86)	21 (53)	0.008
Age (years)	48.5 ± 10	54.6 ± 14	52.4 ± 14	0.262
Body mass index (kg/m²)	23.9 ± 3	44.9 ± 9 *^,†^	25.1 ± 4	<0.001
Body fat percentage (%)	27.5 ± 11	51.4 ± 11 *^,†^	29.5 ± 8	<0.001
Resting energy expenditure (Kcal/day)	1467 ± 165	1943 ± 398 ^+,^*	1691 ± 327	<0.001
REE/BEE (%)	98.7 ± 8	101.9 ± 18 ^‡^	110.3 ± 12 ^•^	0.007
Systolic blood pressure (mmHg)	114.7 ± 15	132.9 ± 17 ^‡^	131.7 ± 18 ^•^	0.002
Diastolic blood pressure (mmHg)	71.3 ± 15	78.8 ± 10	75.9 ± 11	0.106
Mean arterial pressure (mmHg)	85.7 ± 12	96.8 ± 11 ^‡^	94.8 ± 13 ^•^	0.013
Pulse pressure (mmHg)	50.1 ± 9	54.0 ± 10	55.7 ± 12	0.439
Fasting blood glucose (mmol/L)	4.2 (0.9)	6.0 (3) ***	5.8 (2.1) °	<0.001
HbA1c (mmol/mol)	34.0 (6)	44.0 (25) *	42.0 (16) ^•^	<0.001
Insulin (mIU/L)	6.1 (4)	16.3 (9) ^‡^	3.7 (3) ^•,†^	<0.001
HOMA-IR	1.2 (1)	4.9 (7) ^†,^*	0.95 (1)	<0.001
Total cholesterol (mmol/L)	5.1 ± 0.8	4.9 ± 1	4.6 ± 1	0.180
HDL cholesterol (mmol/L)	1.6 ± 0.4	1.1 ± 0.3 ^+,^*	1.5 ± 0,5	<0.001
LDL cholesterol (mmol/L)	3 ± 0.7	2.9 ± 0.9	2.6 ± 1	0.144
Triglycerides (mmol/L)	1.0 (0.4)	1.7 (0.4) *^,†^	0.9 (0.7)	<0.001
Fibroblast growth factor 21 (pg/mL)	107.9 (116)	160.4 (180) *	174.2 (283) °	<0.001
Hypertension (*n*, %)	-	18 (86)	26 (65)	0.185
Diabetes Mellitus (*n*, %)	-	11 (52)	13 (33)	0.757
Obesity (*n*, %)	-	21 (100)	6 (15)	-
Dyslipidemia (*n*, %)	-	18 (86)	23 (58)	0.718
Metabolic syndrome (*n*, %)	-	19 (90)	20 (50)	0.118
Use of PAD (*n*, %)	-	10 (48)	10 (26)	0.025
Use of insulin (*n*, %)	-	4 (19)	3 (8)	-
Use of statins (*n*, %)	-	10 (48)	16 (40)	0.863
Number of antihypertensives (*n*)	-	2.38 ± 1.5	1.85 ± 1.1	0.038
Use of alpha blockers (*n*, %)	-	4 (19)	28 (70)	-
Use of beta blockers (*n*, %)	-	12 (57)	14 (35)	0.017

* <0.001 for Obese vs. Controls; ^†^ <0.001 for Obese vs. PPGL; ^+^ <0.05 for Obese vs. PPGL; ^‡^ <0.05 for Obese vs. Controls; ^•^ <0.05 for PPGL vs. Controls; ° <0.001 for PPGL vs. Controls. Abbreviations: PPGL, pheochromocytoma; REE, resting energy expenditure; BEE, basal energy expenditure; HbA1c, glycated hemoglobin; HOMA-IR, homeostasis model assessment of insulin resistance; HDL, high-density lipoprotein; LDL, low-density lipoprotein; PAD, peroral antidiabetics.

**Table 2 cancers-11-00485-t002:** PPGL patients before and after surgery.

Factors	Before *n* = 27	After *n* = 27	*p*
Female (*n*, %)	15 (56)	15 (56)	-
Age (years)	51.9 ± 13	53.0 ± 13	<0.001
Body mass index (kg/m²)	24.7 ± 3	26.2 ± 4	<0.001
Body fat percentage (%)	29.3 ± 9	32.3 ± 9	0.034
BEE (Kcal/day)	1509 ± 252	1543 ± 266	0.001
REE (Kcal/day)	1655 ± 311	1477 ± 216	<0.001
REE/BEE (%)	110.8 ± 12	96.5 ± 7	<0.001
Systolic BP (mmHg)	131.8 ± 19	124.9 ± 17	0.084
Diastolic BP (mmHg)	75.7 ± 11	74.5 ± 11	0.598
MAP (mmHg)	94.4 ± 13	91.3 ± 12	0.253
Pulse pressure (mmHg)	56.11 ± 13	55.6 ± 19	0.888
FBG (mmol/L)	5.7 (1.7)	4.8 (0.8)	<0.001
HbA1c (mmol/mol)	42.0 (17)	40.0 (6)	0.018
Insulin (mIU/L)	3.3 ± 3	2.8 ± 3	<0.001
HOMA-IR	0.9 ± 0.9	0.6 ± 0.7	<0.001
Total cholesterol (mmol/L)	4.8 ± 1	4.6 ± 1.1	0.436
HDLc (mmol/L)	1.5 ± 0.5	1.4 ± 0.8	0.237
LDLc (mmol/L)	2.7 ± 1	2.7 ± 0.8	0.771
Triglycerides (mmol/L)	0.8 (0.6)	1.2 (0.7)	0.017
FGF21 (pg/mL)	176.4 (284)	131.3 (225)	<0.001
P-Metanephrine (nmol/L)	3.1 (9)	0.17 (0.2)	<0.001
P-Normetanephrine (nmol/L)	11.6 (14)	0.33 (0.4)	<0.001
Chromogranine (ng/mL)	334.8 (489)	39.6 (38)	<0.001
Hypertension (%)	19 (70)	6 (22)	0.071
Diabetes mellitus (%)	9 (33)	2 (7)	0.037
Obesity (*n*, %)	2 (7)	2 (7)	1.000
Dyslipidemia (*n*, %)	17 (63)	18 (67)	0.775
MS (*n*, %)	14 (52)	6 (22)	0.241
Use of PAD (*n*, %)	5 (19)	2 (7)	0.203
Use of insulin (*n*, %)	3 (11)	0 (0)	0.074
Use of statins (*n*, %)	11 (41)	13 (48)	0.583
Use of AHT (*n*)	1.93 ± 1	0.37 ± 0.7	<0.001

Abbreviations: PPGL, pheochromocytoma/paraganglioma; REE, resting energy expenditure; BEE, basal energy expenditure; BP, blood pressure; MAP, mean arterial pressure; FBG, fasting blood glucose; HbA1c; glycated hemoglobin; HOMA-IR, homeostasis model assessment of insulin resistance; HDLc, high-density lipoprotein cholesterol; LDLc, low-density lipoprotein cholesterol; FGF21, fibroblast growth factor 21; P-, plasma; MS; metabolic syndrome; PAD, peroral antidiabetics; AHT, antihypertensives.

**Table 3 cancers-11-00485-t003:** Selected factors associated with serum FGF21 levels in PPGL patients.

Factors	PPGL (*n* = 40)	
	*r*	*p*
Age	0.435	0.005
Weight	0.267	0.095
Body mass index	0.348	0.028
P-Metanephrine	0.212	0.194
P-Normetanephrine	0.086	0.602
Respiratory Quotient	−0.121	0.474
BEE (Kcal/day)	0.169	0.316
REE (Kcal/day)	0.163	0.336
REE/BEE (%)	0.018	0.915
Systolic blood pressure	0.194	0.231
Diastolic blood pressure	−0.047	0.755
Mean arterial pressure	0.058	0.721
Pulse pressure	0.338	0.032
Fasting blood glucose	0.459	0.002
HbA1c	0.426	0.014
Insulin	0.097	0.551
HOMA-IR	0.248	0.121
Total cholesterol	−0.045	0.785
HDL cholesterol	0.009	0.593
LDL cholesterol	−0.211	0.196
Triglycerides	0.255	0.113

Abbreviations: PPGL, pheochromocytoma/paraganglioma; P-, plasma; REE, resting energy expenditure; HbA1c, glycated hemoglobin; HOMA-IR, homeostasis model assessment of insulin resistance; HDL, high-density lipoprotein; LDL, low-density lipoprotein.

**Table 4 cancers-11-00485-t004:** FGF21 levels in PPGL patients with metabolic syndrome and its components.

Category	Occurrence	*n*	FGF21	*p*
Dyslipidemia	Yes	23	264.9 (343)	<0.001
	No	17	133.7 (102)	
Diabetes mellitus	Yes	13	438.2 (337)	0.001
	No	27	158.0 (170)	
Central Obesity	Yes	6	214.2 (326)	0.486
	No	34	167.1 (280)	
Metabolic syndrome	Yes	20	377.9 (333)	<0.001
	No	20	147.8 (94)	
Hypertension	Yes	26	214.2 (348)	0.085
	No	14	160.4 (131)	

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
