# Peer review of "FGF21 Levels in Pheochromocytoma/Functional Paraganglioma"

_cancers, 2019, doi:10.3390/cancers11040485_

Round 1

Reviewer 1 Report

In this study Klimova et al. aim to evaluate changes in serum FGF21 and any relationship between serum FGF21 and energy/glucose homeostasis in patients with PPGL and a cohort with tumor and one year post-tumor removal. For comparison, the authors include a cohort of lean control individuals and a cohort of obese individuals, as it has been previously reported that serum FGF21 levels are elevated in obese, diabetic subjects. As PPGL is a fairly rare disease, the patient cohorts seem sufficient in number and the design to include obese subjects was well considered. Additionally, assessed parameters within same subject with tumor and 1 year post tumor removal are novel and well designed. The primary data here, although negative are significant and address a novel hypothesis regarding PPGL; namely that increased serum FGF21 is not associated with the hypermetabolism observed in these patients. Authors have appropriately highlighted limitations of studies including wide range of serum FGF21 in humans, cross sectional nature of study, potential influence of antihypertensive or antidiabetic therapies. Although the data sets presented in this manuscript are novel and relevant to the field; the organization, thoroughness, and thought fullness of the Introduction and Discussion are severely lacking. 1) Introduction lacks many seminal references in field of FGF21 regarding human and animal studies on its metabolic effects which is the basis of the manuscript. Could reference individually or cite reviews on subject matter such as: a) Bondurant LB and Potthoff MJ Annual Review of Nutrition 2018 – review of metabolic actions of FGF21 b) Markan KR et al. Diabetes 2014 – in mice circulating FGF21 is solely derived from the liver c) Fisher FM et al. Genes and Development 2012 – FGF21 causes browning of white adipose in rodents 2) The Discussion lacks focus. The key finding of this study, “We did not find a relationship between FGF21 and hypermetabolic state in PPGL.”, is a negative finding but is novel and relevant as the true physiological role of circulating FGF21 in humans, let alone in PPGL patients which present with hypermetabolism let alone in PPGL patients which present with hypermetabolism, remains under investigation. Additionally, the first 5 paragraphs of Discussion are somewhat adequate to cover this controversial biological role of circulating FGF21 in humans. However, “…we believe that it is the presence of hyperglycemia that stimulates FGF21 production…” is completely unsupported by data presented. What authors data does indicate is, as occurs in human obesity, there is elevated serum FGF21 also in obese, PPGL patients. However, the hypothesis regarding hyperglycemia stimulating FGF21 may be suggested by the authors, but it should be clarified; are they referring to obese, PPGL patients? In fact, there is literature from human studies that insulin rather than glucose increases total and bioactive FGF21 (Samms RJ et al. Journal Clinical Endocrinology Metabolism 2017). Therefore, other literature supporting this hypothesis should be cited or the authors should remove statement from discussion as it is currently unfounded. 3) Lack of discussion regarding low insulin levels in PPGL patients and potential connection to altered metabolic state. Could lack of insulin be underlying or contributing to hypermetabolism of PPGL patients? Why don’t insulin levels normalize in PPGL patients 1 year post tumor removal? Is there literature regarding this? This is an interesting piece of data which is completely overlooked by the authors. 4) The authors set out to test for association between serum FGF21 levels and hypermetabolism of PPGL patients. However, did they ever consider the hypothesis that increased catecholamines could be driving the elevation in serum FGF21 levels? Data seems it would support this as levels increased in patients with tumors but normalize upon tumor removel. Literature supporting would include: a) Liang Q et al. Diabetes 2014 – HPA axis mediates liver-brain crosstalk and FGF21 production 5) Authors solely focused on FGF21 actions on adipose to cause browning in the introduction and in the discussion but do not even consider potential of central actions of FGF21 in mediating this effect nor do they mention any of the animal literature which may support this potential mechanism of action. a) Douris N et al. Endocrinology 2015 – central actions of FGF21 signaling results in browning of adipose through increased sympathetic nerve activation b) Petal R et al. Molecular Endocrinology 2015 – glucocorticoids regulate FGF21 via feed forward mechanism bypassing negative feedback of the HPA axis in mice c) Bondurant LB et al. Cell Metabolism 2017- FGF21 regulates metabolism via adipose-dependent and adipose- independent mechanisms 6) “Our data thus suggest that circulating levels of FGF21 in PPGL rather reflect activity of liver than fatty tissue.” - What exactly supports this conclusion? In fact, the data presented in no way support or negate whether the serum FGF21 is being produced by the liver or the adipose or some other source i.e. muscle? Authors may reference again Markan et al. Diabetes 2014 as in rodents the genetic studies have demonstrated the circulating FGF21 is liver derived. This statement should be removed as it is unfounded. Furthermore, authors do not state which commercial FGF21 ELISA was used for assays and do not recognize importance of FAP or fibroblast activation protein and its action on FGF21 bioactivity vs total FGF21 in humans. Have authors quantified total or bioactive FGF21 (see Sammms RJ et al. 2017). 7) Can they state, in comparison to serum FGF21 levels in PPGL versus obese individuals, are the levels in PPGL most likely biologically significant? Furthermore, discussion regarding “FGF21 resistance”, in humans and animals is still controversial and authors should therefore reference the following: a) Fisher FM et al. Diabetes 2010 b) Hale C et al. Endocrinology 2012 c) Markan KR F1000 2018 Overall, this manuscript addresses a novel hypothesis by assessing if there is a relationship between circulating FGF21 levels and the hypermetabolism noted in PPGL patients. Although the authors determined there was no relationship between serum FGF21 and hypermetabolism this data should not be underemphasized as the function of circulating FGF21 in humans remains under investigation. Furthermore, serum FGF21 is elevated in a number of other pathologies such as obesity, diabetes, non-alcoholic fatty liver disease, mitochondrial myopathies, etc. but the mechanisms underlying these observations remain under investigation. Additionally, the data regarding PPGL and obesity/impaired glucose homeostasis is also novel however; many aspects of the Introduction and Discussion lack clarity, organization and complete lack any scientific support. The manuscript would tremendously benefit from a complete reworking of the organization, elimination of aspects of speculation, and thorough discussion and inclusion of literature from both human and rodent studies in the field throughout the Intro and the Discussion as discussed in above points.

Author Response

Reviewer No. 1

We would like to thank the reviewer for his thoughtful and careful review and for the opportunity to significantly improve the quality of our manuscript. We have tried to incorporate all the suggestions and comments suggested by the reviewer. We believe that the manuscript has been improved and is now satisfactory. Specific responses are below.

1) Introduction lacks many seminal references in field of FGF21 regarding human and animal studies on its metabolic effects which is the basis of the manuscript. Could reference individually or cite reviews on subject matter such as:

a) Bondurant LB and Potthoff MJ Annual Review of Nutrition 2018 – review of metabolic actions of FGF21

b) Markan KR et al. Diabetes 2014 – in mice circulating FGF21 is solely derived from the liver

c) Fisher FM et al. Genes and Development 2012 – FGF21 causes browning of white adipose in rodents

References were incorporated in the Introduction and Discussion of the manuscript.

2) The Discussion lacks focus. The key finding of this study, “We did not find a relationship between FGF21 and hypermetabolic state in PPGL.”, is a negative finding but is novel and relevant as the true physiological role of circulating FGF21 in humans, let alone in PPGL patients which present with hypermetabolism let alone in PPGL patients which present with hypermetabolism, remains under investigation.

Additionally, the first 5 paragraphs of Discussion are somewhat adequate to cover this controversial biological role of circulating FGF21 in humans. However, “…we believe that it is the presence of hyperglycemia that stimulates FGF21 production…” is completely unsupported by data presented. What authors data does indicate is, as occurs in human obesity, there is elevated serum FGF21 also in obese, PPGL patients. However, the hypothesis regarding hyperglycemia stimulating FGF21 may be suggested by the authors, but it should be clarified; are they referring to obese, PPGL patients? In fact, there is literature from human studies that insulin rather than glucose increases total and bioactive FGF21 (Samms RJ et al. Journal Clinical Endocrinology Metabolism 2017). Therefore, other literature supporting this hypothesis should be cited or the authors should remove statement from discussion as it is currently unfounded.

We agree with your opinion and have completely revised the sixth paragraph. Text changes are highlighted in the manuscript.

3) Lack of discussion regarding low insulin levels in PPGL patients and potential connection to altered metabolic state. Could lack of insulin be underlying or contributing to hypermetabolism of PPGL patients? Why don’t insulin levels normalize in PPGL patients 1 year post tumor removal? Is there literature regarding this? This is an interesting piece of data which is completely overlooked by the authors.

We have included in the Discussion the current knowledge of changes in insulin levels in patients with PPGL.

4) The authors set out to test for association between serum FGF21 levels and hypermetabolism of PPGL patients. However, did they ever consider the hypothesis that increased catecholamines could be driving the elevation in serum FGF21 levels? Data seems it would support this as levels increased in patients with tumors but normalize upon tumor removel. Literature supporting would include: a) Liang Q et al. Diabetes 2014 – HPA axis mediates liver-brain crosstalk and FGF21 production

We included the hypothesis in the text of the manuscript. On the other hand, the recommended literature rather suggests the influence of the glucocorticoid axis and not the adrenergic one. According to authors, central injection of rmFGF21 did not increase the levels of noradrenaline or adrenaline in the liver, suggesting that sympathetic nerves are not involved in FGF21-mediated hepatic glucose production.

5) Authors solely focused on FGF21 actions on adipose to cause browning in the introduction and in the discussion but do not even consider potential of central actions of FGF21 in mediating this effect nor do they mention any of the animal literature which may support this potential mechanism of action. a) Douris N et al. Endocrinology 2015 – central actions of FGF21 signaling results in browning of adipose through increased sympathetic nerve activation b) Petal R et al. Molecular Endocrinology 2015 – glucocorticoids regulate FGF21 via feed forward mechanism bypassing negative feedback of the HPA axis in mice c) Bondurant LB et al. Cell Metabolism 2017- FGF21 regulates metabolism via adipose-dependent and adipose- independent mechanisms

We incorporated consideration of the potential central action of FGF21 in the Introduction of the manuscript, including references.

6) “Our data thus suggest that circulating levels of FGF21 in PPGL rather reflect activity of liver than fatty tissue.” - What exactly supports this conclusion? In fact, the data presented in no way support or negate whether the serum FGF21 is being produced by the liver or the adipose or some other source i.e. muscle? Authors may reference again Markan et al. Diabetes 2014 as in rodents the genetic studies have demonstrated the circulating FGF21 is liver derived. This statement should be removed as it is unfounded. Furthermore, authors do not state which commercial FGF21 ELISA was used for assays and do not recognize importance of FAP or fibroblast activation protein and its action on FGF21 bioactivity vs total FGF21 in humans. Have authors quantified total or bioactive FGF21 (see Sammms RJ et al. 2017).

This conclusion was mere speculation and therefore the statement has been removed. We have changed the Discussion, using the suggested quotations. In our study, the total FGF21 was measured by a commercial ELISA kit (BioVendor, Modrice, Czech Republic).

7) Can they state, in comparison to serum FGF21 levels in PPGL versus obese individuals, are the levels in PPGL most likely biologically significant? Furthermore, discussion regarding “FGF21 resistance”, in humans and animals is still controversial and authors should therefore reference the following:

a) Fisher FM et al. Diabetes 2010

b) Hale C et al. Endocrinology 2012

c) Markan KR F1000 2018

The text was modified according to the recommendations, including citations.

Yours sincerely,

Dr. Judita Klímová, on behalf of all authors

Reviewer 2 Report

Dear Authors, 

The manuscript of Klimova et al. entitled:  “FGF-21 levels are elevated in pheochromocytoma and diminish after tumor removal” compare the FGF-21 levels of patients with PPGL (before and after surgery) with controls and obese patients. This is a very interesting and original issue with scarce data in the literature so far. 

The major comments to the Authors concern mainly the following points:

-      It isn’t clear to me if all PPGLs were functional. The definition of adrenergic and noradrenergic phenotypes in the methods (lines 234-239) doesn’t clarify if patiengs had pathological catecholamines secretion based on classical definition (x3-4 upper limit of the normal in the urines). This is an important issue because catecholamines overproduction leads to an increase in the resting energy expenditure. 

-      I don’t understand why patients with familial, bilateral or malignant PPGL were excluded. (lines 203-204)

-      I think that it would be interesting to present data about the FGF-21 expression in tissue level (paraffin blocks or frozen tissues? ) 

Kind regards

Anna Angelousi

Author Response

Reviewer No. 2

Thank you for your comment and suggestions, which were incorporated in the new version of our manuscript. We would like to clarify some details.

It isn’t clear to me if all PPGLs were functional.

All PPGLs were functional. We added this information to the manuscript. The values of free plasmatic metanephrines are shown below. The red line represents the upper cut-off value for both metanephrines used in our laboratory.

The definition of adrenergic and noradrenergic phenotypes in the methods (lines 234-239) doesn’t clarify if patients had pathological catecholamines secretion based on classical definition (x3-4 upper limit of the normal in the urines).

The definition of phenotypes is based on plasma metanephrine levels according to Eisenhofer et al. (Pheochromocytoma catecholamine phenotypes and prediction of tumor size and location by use of plasma-free metanephrines. Clin Chem 2005, 51, 735-744). All patients had elevated plasma metanephrines (metanephrine and/or normetanephrine), more than 3 times above the upper reference range.

 I don’t understand why patients with familiar, bilateral or malignant PPGL were excluded.

This was a typographic error. Familial, bilateral and malignant PPGL were excluded after operation. This is one of the reasons why the number of patients after tumor removal is smaller than before operation.  Patients with malignant PPGL were excluded, because of remaining catecholamine-secreting tumor cells after operation. Catecholamines are produced in a smaller amount after operation, but the catecholamine excess is still present. Patients with familial PPGL were excluded because of the unknown, but possible influence of germline mutations on the pathophysiological mechanisms, which we are studying. Patients with bilateral PPGL were mostly patients with germline mutations and post-bilateral adrenalectomy. Bilateral adrenalectomy, albeit adrenal sparing, can influence results after surgery.

 I think that it would be interesting to present data about the FGF-21 expression in tissue level (paraffin blocks or frozen tissues?)

We are working on this data.

Yours sincerely,

Dr. Judita Klímová, on behalf of all authors

(graphs are in PDF version)

Reviewer 3 Report

Title: FGF-21 levels are elevated in pheochromocytoma and diminish after tumor removal

This Study provides further investigation of previously described elevation of FGF21 expression in visceral adipose tissue from pheochromocytoma patients.  As an improvement of the previous results, the author assessed circulation FGF-21 levels instead, and furthermore compared the concentration before and after tumor removal.

There some few aspects that need to be clarified before it can be accepted for publication

Pheochromocytoma-induced BAT has been well known as the mechanism of elevated FGF-21. The Author showed, however, no difference in Body Fat percentage in Table 2. This of course does not directly reflect the changes in BAT. The Author need to add this information in the result section, otherwise the result will remain less convincing.

Line 189-190: this statement made the whole study seemingly irrelevant as the main aim of this was actually to correlate the effect of catecholamine excess on FGF 21 level.

As the Body Fat percentage did not change, the best control to be included would be Patients with non-functional Adrenal Tumor with follow ups before and after tumor removal. Would it be possible to obtain such group in the Author’s study center?

The Title needs to be re-formulated as the study population included PPGL patients. Moreover, it has to be made clear in the manuscript how many of the patient had Adrenal and Extra-adrenal Pheo.

Again the Terminology Adrenalectomy is inappropriate since the study population included extra-adrenal  Pheo  cases

The Author needs to provide data that Hypercortisolismus has been ruled out in the Obese control.

15% of PPGL patients were obese; those should not have been included in this group.

 17 Patient did not eventually undergo Operation, any explanation regarding this?

There were 20 patients with metabolic syndrome in table 4, however only 13 with Diabetes, the Author needs to clarify this

It has to be mentioned that no other Syndrom occurred within the PPGL population. Were MEN-syndrome for example ruled out?

Line 175-176: please explain the meaning of “weak link …” since there were no data on tables available regarding the Pheo’s type (adrenergic  vs noradrenergic)

Author Response

Reviewer No. 3

Thank you for your review and suggestions, which were incorporated in the new version of our manuscript. We would like to clarify some details.

Line 189-190: this statement made the whole study seemingly irrelevant as the main aim of this was actually to correlate the effect of catecholamine excess on FGF 21 level.

We did not find a correlation between plasma FGF-21 levels and plasma-free metanephrines in our study. But we cannot exclude any paracrine actions and relationships arising from them. We hope that future study of the local expression of FGF-21 will help us to definitively answer this question.

As the Body Fat percentage did not change, the best control to be included would be Patients with non-functional Adrenal Tumor with follow ups before and after tumor removal. Would it be possible to obtain such group in the Author’s study center?

We agree with this statement. This group would unfortunately be very small. The number of patients with non-functional adrenal tumor indicated for adrenalectomy is very low in our study center.   

The Title needs to be re-formulated as the study population included PPGL patients. Moreover, it has to be made clear in the manuscript how many of the patient had Adrenal and Extra-adrenal Pheo.

Our group of PPGL included 38 pheochromocytoma (22 right and 16 left) and 2 paraganglioma. We included this detail in the manuscript.

Again the Terminology Adrenalectomy is inappropriate since the study population included extra-adrenal Pheo cases.

The term was replaced by “tumor removal” in the new version of the manuscript.

The Author needs to provide data that Hypercortisolismus has been ruled out in the Obese control.

Obese patients are dispensarized in our Obesity and Diabetes Clinical Center. Exclusion of endocrine causes of obesity is one of the basic examinations at our clinic. All had hypercortisolism excluded. We have included this information in the text of the manuscript.

15% of PPGl patients were obese; those should not have been included in this group

It would be troublesome to exclude all obese patients in this group. Weight changes were distinct after operation in some patients (for example +15 kg), so some patients moved into the “Obese” category after operation.

Data excluding those obese PPGL patients can be seen below. There is no distinct change in the results with or without obesity.

Table 1 (in PDF version): Group of PPGL patients without (n=34) and with obesity (n=40) 

Table 2 (in PDF version): FGF-21 plasma levels and its correlation in 34 PPGL patients without obesity

17 Patient did not eventually undergo Operation, any explanation regarding this?

It is a consecutive sample of patients. Some patients did not fulfil the requirements of time interval after operation. We also excluded several patients in the “after operation group” because of newly discovered malignant generalization and genetic testing results (familial PPGL).  

There were 20 patients with metabolic syndrome in table 4, however only 13 with Diabetes, the Author needs to clarify this.

We used the definition of metabolic syndrome which also involves impaired glucose tolerance and impaired fasting glucose, but the Table includes only diabetic for the purposes of clarity.

It has to be mentioned that no other Syndrom occurred within the PPGL population. Were MEN-syndrome for example ruled out?

Yes, familial PPGL was ruled out in the PPGL “after operation” group.

Line 175-176: please explain the meaning of “weak link …” since there were no data on tables available regarding the Pheo’s type (adrenergic  vs noradrenergic)

This sentence referred to lines 133-134.  FGF-21 levels differed in PPGL with adrenergic and noradrenergic phenotype, but were slightly above the level of significance (p = 0.062).

(Graph in PDF version)

Yours sincerely,

Dr. Judita Klímová, on behalf of all authors

Round 2

Reviewer 3 Report

The Authors have adressed my concern. I am satisfied with the Answers.